# Magnesium Silicate Binding Materials Formed from Heat-Treated Serpentine-Group Minerals and Aqueous Solutions: Structural Features, Acid-Neutralizing Capacity, and Strength Properties

**DOI:** 10.3390/ma15248785

**Published:** 2022-12-08

**Authors:** Tatiana K. Ivanova, Irina P. Kremenetskaya, Valentina V. Marchevskaya, Marina V. Slukovskaya, Svetlana V. Drogobuzhskaya

**Affiliations:** 1I.V. Tananaev Institute of Chemistry and Technology of Rare Elements and Mineral Raw Materials, Kola Science Centre, Russian Academy of Sciences, 184209 Apatity, Russia; 2Laboratory of Nature-inspired Technologies and Environmental Safety of the Arctic, Kola Science Centre, Russian Academy of Sciences, 184209 Apatity, Russia; 3Mining Institute, Kola Science Centre, Russian Academy of Sciences, 184209 Apatity, Russia

**Keywords:** serpentine, heat treatment, antigorite, lizardite, chrysotile, hydrate phases, compressive strength, X-ray diffraction analysis, differential scanning calorimetry, acid-neutralizing capacity

## Abstract

The influence of structural features of three serpentine-group minerals (antigorite, chrysotile, and lizardite) on the hydration of heat-treated materials and the formation of magnesium silicate binder has been studied. Initial serpentine samples have been fired in the interval 550–800 °C with a step of 50 °C; acid neutralization capacity (ANC) values have been determined for all samples. Antigorite samples (SAP) have exhibited a maximum reactivity at a temperature of 700 °C (ANC 7.7 meq/g). We have established that the acid-neutralizing capacity of chrysotile and lizardite samples in the temperature range of 650–700 °C differ slightly; the capacity varied in the interval of 19.6–19.7 meq/g and 19.6–19.7 meq/g, respectively. The samples obtained at optimal temperatures (antigorite—700 °C, lizardite, and chrysotile—650 °C) have been studied. Heat-treated serpentines have interacted with water vapor for a year; serpentine hydration has been investigated. The strength characteristics of the resulting binder agents were studied after 7, 28, 180, and 360 days. Upon hardening within 7 days, the strengths of the SAP and SCH samples have been almost the same (2.2 MPa), whereas this indicator for the SLH and SLK samples has been significantly lower (0.5 MPa). After hardening for over a year, the chrysotile sample SCH had the highest strength (about 8 MPa), whereas the strength of antigorite SAP was 3 MPa. The samples of initial, heat-treated, and hydrated heat-treated serpentines have been studied using XRD, differential scanning calorimetry, and surface texture analysis. The serpentine structure is crucial in destroying the mineral crystal lattice during heat treatment. In contrast to heat-treated chrysotile and lizardite, antigorite did not adsorb water. Structural features of chrysotile provided the highest compressive strength of the binding agent compared with antigorite and lizardite. The acid-neutralizing ability of lizardite was noticeably higher than antigorite, whereas its compressive strength was lower due to the layered mineral structure and impurities. We have established that the minerals’ structural features are crucial for the hydration of heat-treated serpentines; the structure determines material utilization in various environmental technologies.

## 1. Introduction

The formation of magnesium silica hydrate (M-S-H) gel and its structural characterization, along with its binding attributes, was reported in multiple studies [1,2,3]. M-S-H-based materials are the class of alternate binders; they were initially considered for their improved refractory performance and the lower pH of their pore solution [4]. Due to a cell lattice-type crystal structure and small sizes of the particles, the water demand for a workable magnesium oxide (MgO)-based binder is significantly higher than that of traditional calcium silicate binder formulations [3]. The higher water demand prevents the binder from achieving a compressive strength comparable to that of the calcium silicate system.

Magnesium silicate binder is usually made from magnesium oxide and silica. Metakaolin is a highly reactive aluminosilicate. It can replace silica fume in the binder system based on M-S-H, considered in the work [4]. Serpentine minerals (3MgO_2_·2SiO_2_·2H_2_O) can also make a magnesium silicate binder [5].

The binding properties of heat-treated (thermally activated) serpentine minerals were actively studied in the 1950s [6]. The material obtained based on dehydrated serpentinite is called serpentinite cement [7]. It was found that in a certain temperature range, a metastable phase of metaserpentine MgO·SiO_2_ is formed as a result of serpentine minerals dehydration [8]. This phase is why the materials based on thermally activated serpentine minerals demonstrate binding properties.

On the other hand, thermally activated serpentines in the form of magnesia-silicate reagents were proposed for water purification from heavy metals [9]. The influence of the serpentine minerals structure on the reagent activity was considered in [10]. A method for determining the optimal firing conditions was proposed in [11]. The features of interaction of the reagent with heavy metals solutions were determined in [12]. The most effective pre-treatment for serpentine was determined to be thermal activation [13]; mechanical activation is also a suitable method. Mechanically activated serpentine minerals can purify solutions from metals, such as copper [14].

The magnesium silicate reagent’s ability to reduce the acidity of solutions is caused by products of the destruction of the original serpentine mineral, mainly magnesium oxide. The alkaline properties of heat-treated serpentines are caused by the formation of a metastable phase. The phase interacts with solutions as a mixture of active oxides of magnesium and silicon [15]. The active phase forms during the destruction of the structure as a result of the dehydration of serpentine minerals. The firing temperature should meet the following condition: hydroxyl water should leave crystal lattices without forming new mineral phases [16].

A unique feature of serpentine-group minerals is their wide variety; the group includes numerous crystalline modifications [17]. Serpentine minerals can be assigned to one of three structural groups (antigorite, chrysotile, or lizardite). The group depends on the compensation method for mismatch in the sizes of the unit cells of the tetrahedral and octahedral layers. Serpentinites studied in the present work contained minerals of various crystalline modifications.

This paper investigates serpentine products suitable for obtaining magnesia-silicate reagents. Mining waste contains these serpentines; their wide application will contribute to the solution of two problems—the utilization of serpentine-containing wastes and the purification of aqueous solutions from metals. Granular magnesium silicate reagent made from thermally activated serpentine minerals is a promising material for purifying highly concentrated solutions. Astringent properties are necessary to obtain granular materials; the properties will differ significantly for serpentines with different structures.

The conditions for the serpentines’ thermal activation are determined in accordance with the requirement to obtain materials with the highest possible degree of transformation of a serpentine mineral into an active metastable phase. Since the type of the precursor magnesia-silicate binder directly affects the quality and properties of the final product [5], the purpose of this work is to study the effect of the structure of the initial serpentine minerals on the hydration process of thermally activated serpentines and the strength of the binder based on them.

## 2. Materials and Methods

### 2.1. Materials

#### 2.1.1. Characteristics of Initial Materials

Serpentinite samples differed in the mineral structure were studied. The antigorite sample (SAP) was isolated from geological samples from the Pechenga ore field (Murmansk region, Russia) and contained olivine impurity. The chrysotile sample (SCH) was isolated from the overburdened rock of the Khalilovo magnesite deposit (Orenburg region, Russia). Two lizardite samples from deposits from the Murmansk region, Russia, were also studied: an SLH sample from the Khabozero olivinite deposit and an SLK sample with vermiculite and pyroxenes impurities from the Kovdor phlogopite deposit. Table 1 presents the chemical composition of the samples.

#### 2.1.2. Preparation of Heat-Treated Serpentine Samples

The serpentine samples were milled in a ball mill for 1.5 h with a weight ratio of material and grinding bodies of 1:3 and particle size less than 8 μm. Heat-treated serpentine samples were obtained via calcination in a muffle furnace (Nabertherm, Program controller S27, Bremen, Germany). Materials were spread in a thin layer on a metal pan and placed in a furnace heated to a preset temperature; the firing time was 20 min. The firing temperature changed in the interval 550–800 °C with a step of 50 °C.

### 2.2. Experimental Methods

#### 2.2.1. Determination of Serpentines’ Acid Neutralization Capacity

The acid-neutralizing capacity of serpentine samples was determined by acidimetric titration [18]. The citric acid method of determining MgO powder reactivity was used, for example, in work [19]. We used a 0.02 N hydrochloric acid (HCl) solution where a portion of serpentinite (250 mg) was placed into a tube with 250 mL of 0.02 N HCl. The suspension was stirred for 3 h and then rested for 24 h for the acid solution’s partial neutralization. Then the suspension was filtered through a paper filter with a pore diameter of 1–2 μm, and the acid remaining in the solution was titrated with a 0.01 N Na_2_CO_3_; methyl red was used as an indicator. The acid-neutralizing capacity (reactivity) of heat-treated serpentines was calculated as the difference between the initial and remaining acid amounts [20]; the formula for calculating ANC is given below. The result obtained was converted to the calcined residue (considering LOI 1000). The conversion to the calcined residue makes it possible to compare samples with different amounts of water, which remains in the structure of the serpentine mineral after calcination.
$$\mathrm{ANC}=\frac{\left(C_{0}-C_{1}\right)\cdot V\cdot 1000}{m}\cdot \frac{100}{\left(100-\mathrm{LOI}\right)},$$

ANC—acid-neutralizing capacity, meq/gC_0_—hydrochloric acid solution initial concentration, mol/LC_1_—hydrochloric acid solution concentration after interaction with serpentine, mol/Lm—serpentine sample portion, gV—volume of acid solution, LLOI—loss on ignition at 1000 °C, wt.%.

#### 2.2.2. Study of the Active Serpentine Phase Formation

To assess the ability of serpentines to form an active phase, the portion of HCl 8 wt.% (0.4 L) was heated in a thermostat up to 90 °C, then serpentine samples (20 g) were added. The suspension was soaked for 5 min under stirring and filtered through white ribbon filter paper. The solutions and dried precipitates were analyzed for silicon and magnesium content by atomic emission spectrometry.

#### 2.2.3. Study of the Hydration of Heat-Treated Serpentine Samples

The interaction of heat-treated serpentine samples with water vapors was studied. Separately weighed amounts of heat-treated serpentines were placed in the experimental chamber for 1, 11, 60, 180, and 360 days at constant temperatures of 20 ± 2 °C and relative humidity of 95% and 75%. The LOI of materials was determined.

#### 2.2.4. Study of the Binding Properties of Heat-Treated Serpentine Samples

The binding properties of heat-treated serpentine samples were studied through their interaction with water. The samples of heat-treated serpentinites were mixed with water, and a plastic clay-like mixture was obtained. The water demand of powders (solid/liquid ratio) for the plastic mixture obtained ranged from 0.37 (SLK, SAP) to 0.40 (SLH) and 0.44 (SCH), which corresponds to the normal density of cement grout. Molded into cubic with a rib size of 1.41 mm, samples were solidified at a temperature of 20 ± 2 °C and relative humidity of 90–95% during 7, 28, 180, and 360 days. The obtained samples were used to study the hydration of heat-treated serpentines. The strength of these samples was determined using the PGM-100MG4A press (LLC “Special Design Bureau Stroypribor”, Chelyabinsk, Russia).

### 2.3. Physico-Chemical Methods

The transformations of the samples upon heating were studied by the method of thermogravimetry (TG) and differential scanning calorimetry (DSC) using a thermal analyzer STA409 PC Luxx NETZSCH (Netzsch-Gerätebau GmbH, Selb, Germany). The samples were heated in air at a heating rate of 10 °C/min.

X-ray diffraction (XRD) analyses of the antigorite and chrysotile samples were performed using a powder diffractometer Shimadzu XRD-6000 with CuKα radiation (Shimadzu Corporation, Kyoto, Japan). Scanning was carried out in a 2θ deg range of 6 to 70°, with the step of 0.02° and a dwell time of 1 s. Phases were determined using a PDF-4+2021 program with ICDD’s integrated data-mining software (International Centre for Diffraction Data, Newtown Square, PA, USA). XRD studies of the lizardite samples were performed with an automatic powder X-ray diffractometer Bruker Phaser D2 using CoKα radiation (Bruker Corporation, Karlsruhe, Germany). Analyses were conducted in a 2θ deg range of 6 to 70°, with a step size of 0.01° and a measuring time of 1 s per step. Phases and crystallinity were determined using a PDF-2 powder database with DIFFRAC.EVA program, pattern numbers: antigorite 04-015-2964, lizardite 00-50-1625, chrysotile 00-052-1563, forsterite 01-074-1683, enstatite 01-082-3779, vermiculite 00-060-0341, hematite 01-079-0007, olivine 01-087-2044, pyroxene (enstatite) 04-021-7224.

The infrared (IR) spectra were recorded on Spectrum Two (PerkinElmer, Inc., Waltham, MA, USA) and were collected in transmittance mode from 4000 to 450 cm^−1^ at a resolution of 1 cm^−1^. Micromorphological research was conducted with the digital scanning electronic microscope SEM LEO-420 (Carl Zeiss AG, Oberkochen, Germany).

X-ray fluorescence spectrometer Spectroscan Max-GV (SPA Spectron, Saint-Petersburg, Russia) was used to determine the chemical composition of serpentine samples. The concentrations of the components in the solution were determined by atomic emission spectrometry on an ICPE-9000 instrument (Shimadzu, Kyoto, Japan).

## 3. Results

### 3.1. Characteristics of Heat-Treated Serpentines

The temperature interval for forming an active metastable phase was in the range of 550–800 °C, according to the data of thermal analysis of serpentines (Figure 1). The dependence of activity on temperature is extreme since it is determined by the completeness of two processes that affect this indicator differently. On DSC curves, these processes correspond to the endothermic effect of the crystal lattice destruction of the serpentine mineral and the exothermic effect of the formation of high-temperature phases. The superposition of two peaks on DSC curves of the antigorite sample was observed, i.e., the formation of high-temperature inactive compounds began even before the completion of the serpentine lattice destruction (Figure 1a, curve 1). In contrast, the temperature of these processes in lizardite and chrysotile differed by about 100 degrees (Figure 1b–d, curve 1), i.e., the formation of high-temperature phases began after the destruction of the initial serpentine minerals.

Breuil et al. [21] demonstrated that magnesium leaching from samples correlates with the content of the active amorphous phase; thus, the acid-neutralizing capacity reflects the interaction activity of heat-treated serpentines with aqueous solutions. This process is a key factor in forming the magnesia-silicate phase [22]. The acid-neutralizing capacity of samples obtained at different firing temperatures is demonstrated in Table 2. Sample SAP (antigorite) exhibited maximum activity at a temperature of 700 °C. The acid-neutralizing capacity of SCH and SLH samples in the temperature range of 650–700 °C differed slightly. Further, the samples obtained at optimal temperatures (antigorite—700 °C, lizardite, and chrysotile—650 °C) were studied.

The reactivity of the samples obtained at the optimal temperature increased in the SAP-SLH-SCH series, which corresponded to the values of the activation energy for the dehydroxylation process. Chrysotile was the least stable serpentine mineral with an activation energy of 184 kJ/mol; this value for lizardite and antigorite was significantly greater—221 and 255 kJ/mol, respectively [23].

Evolution in the serpentine mineral composition after heat treatment was monitored by DSC and XRD. Endo-effect significantly reduced the DSC curves of the heat-treated lizardite and chrysotile caused by dehydroxylation of the serpentine minerals. Therefore, high destruction degree of the original serpentine minerals was observed under optimal firing conditions (650 °C).

XRD patterns had broad humps from 17 to 43° (2Θ) in all heat-treated samples (Figure 2, curve 2), suggesting the presence of an amorphous component [15]. Heat-treated chrysotile (SCH) and lizardite (SLH, SLK) samples did not contain crystallized serpentine minerals. Simultaneously, the formation of magnesium silicates was observed; in particular, micro- to nanocrystal forsterite Mg_2_SiO_4_ and enstatite Mg_2_Si_2_O_6_ in an amorphous phase were found, according to [24].

The IR spectra patterns proved the forsterite presence in a heat-treated SCH sample. Bands at 874 and 991 cm^−1^ corresponded to the vibrations of SiO_4_ tetrahedra in the forsterite structure; the broadening of these bands indicated its amorphous state. Heat-treated SLH and SLK contained micro- to nanocrystal enstatite in an amorphous phase along with forsterite (Figure 2c,d, curve 2). For these samples, the content of the amorphous phase was determined, and the content of crystalline serpentine minerals was also quantified (Table 3). The crystallinity of serpentine concentrates went down after roasting at 650 °C. The content of the crystalline serpentine mineral has mostly decreased in the lizardite SLH from 75.2 to 4.8 wt.%, whereas in the SLK samples from 44.8 to 13.9 wt.%.

Antigorite better retains its original structure under the optimal firing temperature (700 °C). The main endo effect of the crystal lattice destruction of the serpentine mineral was preserved on the DSC curve (Figure 1a, curve 2). Main basal reflexes of the initial mineral were observed for the heat-treated antigorite sample obtained at 700 °C (Figure 2a, curve 2). Antigorite is more resistant to roasting due to its less defective structure when compared with other serpentine varieties [25]. Relatively large separate microblocks in the antigorite structure cause its stability and difficulties of water diffusion from the crystal lattice. Diffusion of water vapor in the antigorite lattice in the direction perpendicular to the basal plane (axis *c*) is absented. Diffusion along the *a*-axis was limited, and only along the *b*-axis did water molecules move freely [23].

The degree of serpentine activation was calculated as the ratio between experimental and theoretical acid-neutralizing capacity (%) (Table 4). Theoretical acid-neutralizing capacity was calculated from the magnesium content in heat-treated samples. The degree of activation was 92% for chrysotile, 72% for lizardite, and 38% for antigorite. The lower the activation energy of dihydroxylation, the higher the degree of activation.

The study of serpentine thermolysis supervised by N.O. Zulumyan has shown that the efficiency of serpentine amorphization is primarily associated with the conditions of their formation in the Earth’s crust [26]. The degree of silica extraction upon acid treatment in the current study was done in accordance with the method proposed by N.O. Zulumyan since it is the most objective index of the transformation degree of serpentine minerals into meta-serpentine [27]. Heat-treated serpentine leaching by HCl (8 wt.%) led to the migration of (SiO_4_)^4−^ anions from the serpentine silicate layer to the solution as orthosilicic acid [25]. The silicon-containing phase, which was not leached by HCl (8 wt.%), had low activity and participated in the formation of a magnesia-silicate binder only under conditions of the alkaline component (MgO) high concentrations [22]. Silicon was leached to a greater extent from lizardite (34%) and to a lesser extent from antigorite (28%) and chrysotile (26%). This indicator was not related to the activation energy of the dehydroxylation reaction of serpentine minerals and may reflect the impurities effect on the silicate components’ solubility. The silica solubility in the magnesia-silicate system with aluminum was higher [28], and with calcium was lower [29] compared to a magnesia–silicate system without impurities.

Amorphous silica, diagnosed by a reflex with a maximum of 22° (2Θ), was found in the leaching residues of all heat-treated serpentines (Figure 2a–c, curve 4). Additional silicate phases were not diagnosed in the SCH sample; simultaneously, initial serpentine, forsterite, and enstatite were found in the SAP sample; forsterite and enstatite were determined in the SLH sample.

The degree of magnesium leaching by HCl (8 wt.%) was near 100% for chrysotile and 82–84% for lizardite and antigorite. This confirms the absence of magnesium silicates in the chrysotile leaching residue. The magnesium silicate products of the initial serpentines’ destruction in residues of antigorite and lizardite were more resistant to the reaction with hydrochloric acid.

The Mg/Si index in the reacting system is an important indicator affecting the composition and properties of the magnesia-silicate binding agent [22]. This ratio varied from 3.9 to 5.1 in the active phase, which indicated Mg excess, i.e., the mineral formation during the interaction of heat-treated minerals with water proceeded in a system with high alkalinity. The pH values of aqueous suspensions of serpentines (heat-treated/hydrated heat-treated) increased in the series SAP (10.23/9.40)—SLK (10.34/9.55)—SCH (10.4/10.06).

### 3.2. Interaction of Heat-Treated Serpentines with Water Vapors

The thermal destruction of serpentines leads to the formation of magnesium-containing phases prone to hydration [30]. In studies on optimizing the composition of magnesia binders, magnesium oxide hydration is considered a key link in the formation of the strength of magnesia stone [31]. When interacting with water vapors, the amount of adsorbed water depends on the chemical nature of the magnesium-containing phases and the value of the particle’s surface [30,32]. Thus, the kinetic parameters of water sorption by heat-treated serpentine particles should correlate with the acid-neutralizing capacity of the obtained material.

Heat-treated antigorite (SAP) did not adsorb water from the air (Figure 3). This observation confirms that antigorite retained the most ordered structure after heat treatment [23]. Heat-treated lizardite (SLH) and chrysotile (SCH) interacted with water vapors more intensively than heat-treated antigorite. Higher values of water sorption in chrysotile than in lizardite can be explained by the layered structure and the aluminum presence in lizardite, which reduce the reactivity of heat-treated lizardite [33].

### 3.3. The Hydration of Heat-Treated Serpentines

In contrast with the initial heat-treated samples, an additional endothermic effect at a temperature of 110–120 °C corresponding to the removal of physically bound water was revealed on the thermograms of hydrated heat-treated samples obtained after hardening for 28 days under wet conditions (Figure 1, curve 3). At a temperature of 350–600 °C, the indefinite wide peak of the endothermic effect of water removal was observed; the peak is attributed to the dehydration process of both brucite Mg(OH)_2_ and magnesium silicate hydrate (M-S-H) phases of variable composition [34,35]. M-S-H phases were identified by XRD reflections in 2Θ 22, 35, and 60° areas [22].

Wide reflections corresponded to the decomposition products of serpentine minerals; forsterite and/or enstatite were observed in X-ray patterns of heat-treated serpentines in the areas of 2Θ 36–38° and 59–63°. The same compounds were retained in hydrated samples. According to [35,36,37], these phases can be referred to as M-S-H phases, which have binding properties. The mechanism of the M-S-H phase formation during the interaction of heat-treated serpentine (lizardite) with an aqueous solution is described as sequential reactions: silica and magnesium dissolution; silica polymerization and precipitation on the surface of reacting particles; magnesium sorption from the solution to form a magnesium silicate phase [38].

The hydration products can be identified using differential scanning calorimetry (DSC). Based on the results presented in [35,39], the mass losses were determined in the temperature ranges (°C): 20–350 (dehydration, phase D), 350–600 (dehydroxylation of magnesium silicate binder and binder precursor, phase B), 600–900 (dehydroxylation of serpentine minerals, phase S). This indicator was determined for initial, heat-treated, and hydrated heat-treated (after 28 days of hardening) samples. Table 5 shows data on the phase content calculated for a dehydrated sample, i.e., excluding phase D. The general trend for all hydrated heat-treated samples was a decrease in the phase S content compared to initial samples and an increase in phase B compared to heat-treated samples. The number of phases and their ratio related to the degree of activation were calculated by the acid-neutralizing capacity. The greater the degree of activation, the less is phase S; the more phase B is, the greater the relative content of phase B is, calculated as phase B/(phase B + phase S).

### 3.4. Compressive Strength of Binder—Molded Hydrated Heat-Treated Serpentines

The binder formation during the interaction of heat-treated serpentines with water allows one to obtain materials with certain strengths. The compressive strength characteristics of the samples appear in Table 6. Upon hardening within 7 days, the strengths of the SAP and SCH samples were almost the same (2.2 MPa), whereas this indicator for the SLH and SLK samples was significantly lower (0.5 MPa). Upon hardening for over a year, the chrysotile sample SCH had the highest strength (about 8 MPa), whereas the strength of antigorite SAP amounted to 3 MPa. Although the lizardite strength SLH increased to 1.1 MPa after a year of hardening, this indicator remains lower than in other varieties of serpentine. The strength of SLK did not increase over the entire hardening period and remained at 0.5 MPa.

The results presented in Table 6 show the absence of a relation between compressive strength and the content of precursor of magnesium silicate binding agent in the heat-treated serpentines. The content of the heat-treated serpentine active phase increased in the row SAP–SLH–SCH, whereas the strength of the SLH samples was less compared with the SAP samples. Thus, the sample strength was affected not only by the active phase content but also by factors primarily attributed to the structure of serpentine minerals. The layered structure of microcrystallites, which is pretty common for lizardite, is the most likely reason for this discrepancy [40].

Microscopic studies assist in explaining the results of the compressive strength. Unlike the loose microstructure of lizardite (Figure 4a), the particles of the antigorite constitute a uniform, dense material (Figure 4b). The surface texture of the particles of the initial chrysotile was characterized both by a porcelaneous shell-like fracture (as in antigorite) and by the presence of microcrystallinity (as in lizardite) (Figure 4c).

New visible phases formed after the interaction between heat-treated serpentine minerals and water (Figure 4b,d,e). The particles of antigorite and chrysotile made a uniformly compacted homogeneous structure in the newly formed binding material (Figure 4d,f), while the lizardite sample represented a conglomerate of chaotic oriented plates, significantly reducing the strength of the resulting material (Figure 4b).

The strength of the hardening products was affected by the crystalline structure of serpentine minerals as well as by their origin, namely their chemical composition, content, and impurity profile. For instance, the variations in the strength of the SLH and SLK samples in which lizardite is the main serpentine component were observed. After hydration, the crystallinity reduced in the SLH sample and increased slightly in the SLK sample (Table 3). XRD revealed the recrystallization of a serpentine mineral in the SLK sample (Figure 2d, curve 3). A decrease in the amorphous phase from 55.5 in heat-treated SLK to 48.5% in the hydrated heat-treated SLK sample indicated the processes of reverse crystallization of the binder precursor during the formation of binding material, which led to the reduction of the material strength. In addition to the difference between lizardite samples in terms of the content of the serpentine component and its behavior during firing and hydration, the mineral composition of impurities was also noteworthy. Due to its origin, the Kovdor serpentine sample differed from Khabozero one since it contained vermiculite, which negatively affected the strength of target materials.

## 4. Discussion

The key factor for forming the magnesia-silicate phase is the interaction activity of heat-treated serpentines with aqueous solutions. The reactivity of the samples was evaluated by the ANC value. This indicator reflects the ability of thermally activated samples to interact with aqueous solutions and form hydrated phases. The maximum possible ANC value for the studied serpentines is meq/g: 7.7 (SAP), 18.9 (SLH), and 19.7 (SCH). The reactivity of the samples obtained at the optimal temperature increased in the series antigorite– lizardite—chrysotile. The degree of transformation of serpentines into the active metastable phase increases in the same row. Reactivity corresponded to the values of the activation energy for the dehydroxylation process; the lower the activation energy of the dehydroxylation reaction, the higher the ANC.

It has been found that heat-treated antigorite did not sorb water, in contrast to heat-treated chrysotile and lizardite. This fact indicates that the hydration of the antigorite occurs only on the particle’s surface.

Weight losses of hydrated samples at a temperature of 350–600 °C reduced in the same sequence, from chrysotile (6.3 wt.%) to lizardite (4.2 wt.%) and antigorite (3.8 wt.%). Therefore, weight loss could be considered an indirect indicator of the total content of the binding agent and binder precursor formed during the hydration of heat-treated serpentines.

Unlike the binder content, the compressive strength of the samples based on heat-treated serpentines decreased in the range of chrysotile–antigorite–lizardite; compressive strength at 28 days was 3.5, 2.4, and 0.7 MPa, respectively. The structural features of chrysotile determined the greatest strength of binding material compared to antigorite and lizardite. Although the lizardite activity was noticeably greater than the antigorite, its strength was less due to the layered mineral structure and the impurities.

## 5. Conclusions

Serpentinites are common in the Earth’s crust; they form serpentine areas with specific vegetation in certain regions. Waste containing serpentine minerals forms in the development of magnesite, olivinite, sulfide ores, diamonds, and phlogopite deposits [41,42]. Currently, there are various options for the disposal of serpentinites used to produce building materials, slow-release fertilizer, and other materials [43]. However, unlimited reserves of serpentine raw materials stimulated the search for new technology for their processing.

The greatest interest involves the production of materials that can be used to solve some current problems, including those in the field of environmental management. The presented results aim to solve the problem of recycling mining wastes with a high content of serpentine minerals by obtaining a granular magnesium silicate reagent for cleaning solutions with a high content of metals.

The study of the process of formation of magnesium silicate binder based on thermally activated serpentines and water showed that the structural features of serpentines were crucial in the hydration of the heat-treated mineral. As a result, the material selection for producing a granular magnesium silicate reagent is very important.

Chrysotile is suitable for producing granular materials due to the high reactivity of the thermally activated product in the binder formation process, which makes it possible to obtain a granular material with high strength. Lizardite exhibits high activity; however, the layered structure of microcrystallites and the presence of impurities reduces the strength of the binder. Lizardite is advisable to use as fractionated powders [9]. Antigorite differs from the other two serpentines in tits lower acid-neutralizing ability and can be used for obtaining magnesia and silicate products [44].

## Figures and Tables

**Figure 1 materials-15-08785-f001:**
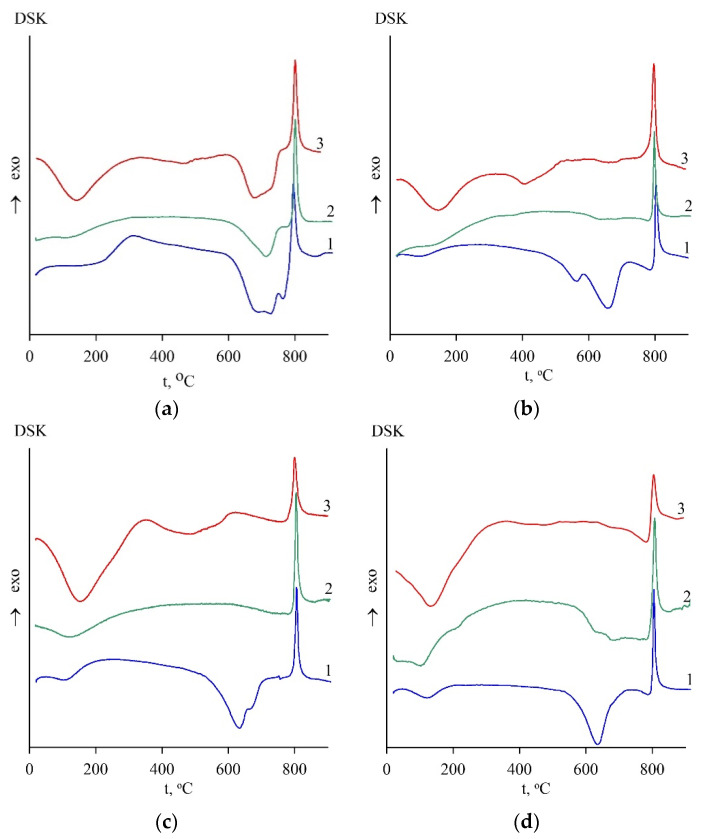
DSC results for serpentine samples: SAP (**a**), SCH (**b**), SLH (**c**), and SLK (**d**); curve for initial (1), heat-treated (2), and hydrated heat-treated (3) samples after 28 days of hardening.

**Figure 2 materials-15-08785-f002:**
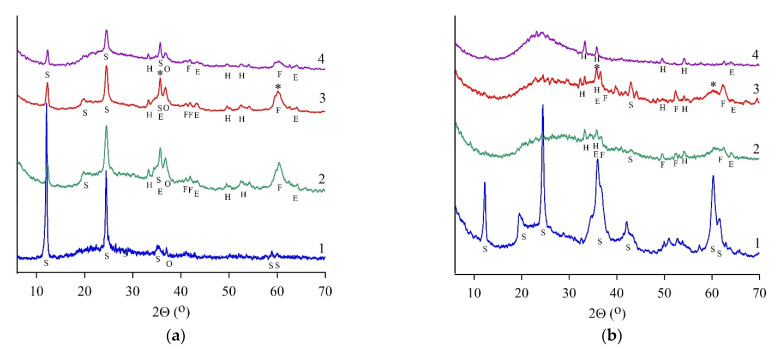
X-ray phase analysis results for serpentine samples: SAP (**a**), SCH (**b**), SLH (**c**), and SLK (**d**); curve for initial (1), heat-treated (2), and hydrated heat-treated (3) samples after 28 days of hardening, (4)—residue after leaching by HCl (8 wt.%). S—serpentine, E—enstatite, F—forsterite, V—vermiculite, H—hematite, O—olivine, P—pyroxene (enstatite), *—magnesium silicate hydrate phase.

**Figure 3 materials-15-08785-f003:**
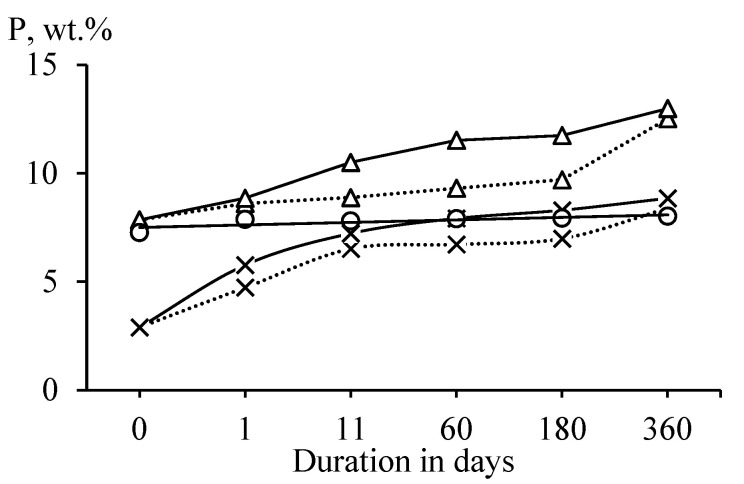
Dependence of the water content P (wt.%) in samples of heat-treated antigorite (○), lizardite (×), and chrysotile (Δ) on the duration of exposure at a temperature of 20 ± 2 °C and a relative score of 95 (^__^) and 75% (‧‧‧‧).

**Figure 4 materials-15-08785-f004:**
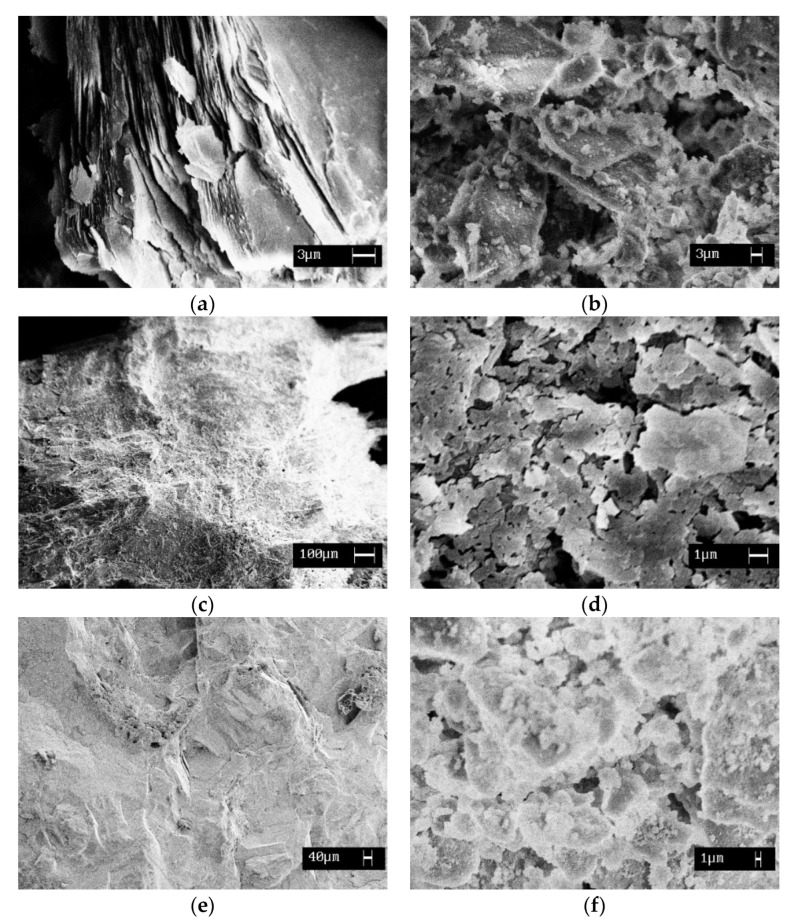
Surface texture of the initial serpentines SLH (**a**), SAP (**c**), SCH (**e**), and binder materials from hydrated heat-treated serpentines SLH (**b**), SAP (**d**), SCH (**f**).

**Table 1 materials-15-08785-t001:** Composition of serpentine samples by XRF (wt.%).

Sample	MgO	SiO_2_	FeO	Fe_2_O_3_	CaO	Al_2_O_3_	LOI	Impurities
SAP	36.0	40.7	5.7	2.2	0.3	0.2	11.8	3.1
SCH	36.3	36.1	1.1	5.4	3.5	1.8	14.6	1.2
SLH	44.9	39.7	-	0.8	1.4	0.4	12.6	0.2
SLK	35.4	39.2	-	1.7	1.8	2.2	17.4	2.3

Note: the dash means that the component content was below the detection limit; impurities included NiO, Cr_2_O_3_, and MnO. LOI—loss on ignition at 1000 °C.

**Table 2 materials-15-08785-t002:** Serpentines’ acid-neutralizing capacity, meq/g.

Sample	Initial	Heat-Treated, °C					
		550	600	650	700	750	800
SAP	1.5	1.7	4.8	7.3	7.7	4.5	-
SCH	8.2	12.6	18.0	19.6	19.7	18.5	9.8
SLH	5.3	12.1	16.9	18.6	18.9	18.4	5.3
SLK	4.0	-	13.3	14.8	15.4	15.1	9.8

**Table 3 materials-15-08785-t003:** Lizardite samples composition (wt.%).

Phase		SLH			SLK		
		Initial	Heat Treated	Hydrated Heat-Treated	Initial	Heat Treated	Hydrated Heat-Treated
Crystalline	serpentine mineral	75.2	4.8	5.6	44.8	13.9	20.1
	other minerals	8.9	42.2	33.9	39.4	30.6	31.4
Amorphous		15.9	53.0	60.5	15.8	55.5	48.5

**Table 4 materials-15-08785-t004:** The degree of serpentine activation.

Sample	Components, (wt.%)			Acid-Neutralizing Capacity			Leaching HCl (8 wt.%)				
	LOI	MgO	SiO_2_	Theoretical, meq/g	Experimental, meq/g	Activation Degree, %	MgO, g/100 g	SiO_2_, g/100 g	Leaching Degree, %		Mg/Si mol
									MgO	SiO	
SAP	2.8	39.7	44.9	20.4	7.7	38	32.6	12.6	82	28	3.9
SCH	7.4	39.5	42.6	21.3	19.6	92	39.0	11.4	98	26	5.1
SLH	8.0	47.9	42.5	26.0	18.6	72	40.0	14.5	84	34	4.1

**Table 5 materials-15-08785-t005:** Thermogravimetry analysis results of serpentine samples.

Sample		Phase B, wt.%	Phase S, wt.%	Phase B/(phase B + phase S), %
SAP	Initial	1.1	12.8	8.0
	Heat-treated	0.7	3.0	18.9
	Hydrated heat-treated	3.8	7.3	34.4
SCH	Initial	4.0	10.0	28.9
	Heat-treated	1.1	3.7	22.1
	Hydrated heat-treated	6.3	2.9	68.4
SLH	Initial	2.3	10.6	17.8
	Heat-treated	1.7	4.6	27.0
	Hydrated heat-treated	4.2	4.2	50.0
SLK	Initial	3.4	9.1	26.9
	Heat-treated	0.7	6.5	9.1
	Hydrated heat-treated	4.8	7.1	40.1

**Table 6 materials-15-08785-t006:** Compressive strength of heat-treated serpentinite samples.

Sample	Compressive Strength Results, MPa			
	7 Days	28 Days	180 Days	360 Days
SAP	2.0	2.4	2.9	3.0
SCH	2.2	3.5	6.6	7.9
SLH	0.5	0.7	1.1	1.1
SLK	0.5	0.5	0.5	-

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
