# Peer review of "Magnesium Silicate Binding Materials Formed from Heat-Treated Serpentine-Group Minerals and Aqueous Solutions: Structural Features, Acid-Neutralizing Capacity, and Strength Properties"

_materials, 2022, doi:10.3390/ma15248785_

Round 1

Reviewer 1 Report

This is an interesting study on M-S-H binders derived from thermally activated serpentine minerals. The information in the manuscript could be better organised. The authors should present all of the characterisation data, i.e., DSC, thermogravimetry, XRD and FTIR results before they discuss the other physicochemical properties of the activated minerals and binders. The fact that they have presented the acid neutralisation capacity (ANC) data before the XRD data means that they cannot fully interpret and explain the ANC results.

The English language and grammar require corrections at the copy editing stage. In particular, the technical and scientific terms (e.g., ‘magnesium silicate cement’ and ‘Portland cement’)  should be used correctly and consistently throughout the manuscript.

It is preferable for the abstract to be written with more explicit detail. As a minimum, the temperature ranges of the heat treatment, selected acid neutralising capacities and the 7- and 28-day compressive strengths should be given.

Lines 51-55: In this section the authors mention some of their own studies without summarising the relevant and interesting information arising from them. The key outcomes of these studies should be stated. What minerals were considered? How does structure influence activity? What were the optimal firing conditions? What were the uptake capacities for the heavy metals? What heavy metals?

Lines 61-63: The authors state, ‘The formation of active phase occurs during the dehydration of serpentine minerals; therefore, the firing temperature should meet the conditions when hydroxyl water loose from crystal lattices without forming new mineral phases [15].’. For completeness, they should also comment on the structural disruption of the crystal lattices during the dehydration process.

Lines 70-74: The objectives section lacks appropriate detail. It would be beneficial for the authors to provide a more explicit account of the aims and objectives to enable readers with existing expertise in this subject to skim-read the methods section and move rapidly on to the results.

The authors must clearly and comprehensively state the novelty of their study at the end of the introduction. They have already carried out several studies in this area (as have others), so they must state what additional/unique contribution the present study makes.

Line 112: The authors should state how the solutions and dried precipitates were analysed.

Section 2.2.4: The exact water:binder ratio for each sample should be tabulated in this section.

Section 3.1: The authors must provide a more comprehensive interpretation of the thermal events depicted in the DSC data shown in Figure 1. They should provide a clear account of all of the observed thermal events prior to making comparisons among the minerals. The data for DSC, thermogravimetry, XRD and FTIR should be presented consecutively to enable these data to be interpreted and explained more comprehensively.

Table 2: The authors should explain the observed trends in acid neutralisation capacity in terms of the composition of the original and heat-treated minerals.

Lines 175-179: This section is confusing. The authors should more fully explain the information given in Lines 175-179. What is meant by ‘activity’? To what reaction do the authors refer when they discuss ‘activation energy’?  

Figure 2: The authors should carry out full refinement on the XRD data for the original and heat-treated mineral samples to give a semiquantitative analysis their phase compositions and total degrees of crystallinity. They must also give the numbers of the powder diffraction files used to identify the phases.

Line 196 onwards: The FTIR spectra are missing. The authors must present and fully interpret the FTIR data.

Table 4: What is the origin of the theoretical acid neutralisation capacity of the minerals given in Table 4?

Table 4: Two columns of figures are given under the heading ‘Leaching degree’. Presumably these correspond with the respective leaching of MgO and SiO2. The table must be corrected to reflect this.

The discussion section is missing from the manuscript. The authors have presented and interpreted their data (in a confusing order), but they have failed to generate a discussion of their findings in the context of existing studies. They should also highlight the technical implications of their findings in the discussion, along with any practical applications of the thermally activated minerals and their binders.

 I have no confidential comments for the editors. 

Author Response

Dear reviewer, 

thank you very much for your valuable comments.

We have corrected the manuscript according to almost all your suggestions.

  1. This is an interesting study on M-S-H binders derived from thermally activated serpentine minerals. The information in the manuscript could be better organised. The authors should present all of the characterisation data, i.e., DSC, thermogravimetry, XRD and FTIR results before they discuss the other physicochemical properties of the activated minerals and binders. The fact that they have presented the acid neutralisation capacity (ANC) data before the XRD data means that they cannot fully interpret and explain the ANC results.

The sequence of material presentation corresponds to the logic of our research: based on DSC, we 1) determined the temperature range at which it makes sense to thermally activate serpentines, then, 2) for all heat-treated samples, we determined the characteristic (acid neutralization capacity, ANC), which is similar to the indicator proposed in the work of Breuil et al. [20]. Instead of magnesium leaching, we measured the samples’ ability to neutralize the HCl solution. 23) Based on the ANC values, we selected the thermal activation temperature and then studied with materials obtained at the optimum temperature.

  1. The English language and grammar require corrections at the copy editing stage. In particular, the technical and scientific terms (e.g., ‘magnesium silicate cement’ and ‘Portland cement’)  should be used correctly and consistently throughout the manuscript.

The terms “magnesium silicate cement” and “Portland cement” have been replaced by “magnesium silicate binder” and “calcium silicate binder”.

English grammar has been checked by native speaker.

  1. It is preferable for the abstract to be written with more explicit detail. As a minimum, the temperature ranges of the heat treatment, selected acid neutralising capacities and the 7- and 28-day compressive strengths should be given.

The abstract has been modified.

  1. Lines 51-55: In this section the authors mention some of their own studies without summarising the relevant and interesting information arising from them. The key outcomes of these studies should be stated. What minerals were considered? How does structure influence activity? What were the optimal firing conditions? What were the uptake capacities for the heavy metals? What heavy metals?

Paragraph Lines 51-55 was expanded, information in accordance with the reviewer recommendations was added.

5  Lines 61-63: The authors state, ‘The formation of active phase occurs during the dehydration of serpentine minerals; therefore, the firing temperature should meet the conditions when hydroxyl water loose from crystal lattices without forming new mineral phases [15].’. For completeness, they should also comment on the structural disruption of the crystal lattices during the dehydration process.

Edited as requested by the reviewer.

  1. Lines 70-74: The objectives section lacks appropriate detail. It would be beneficial for the authors to provide a more explicit account of the aims and objectives to enable readers with existing expertise in this subject to skim-read the methods section and move rapidly on to the results.

This section was expanded.

  1. 7. The authors must clearly and comprehensively state the novelty of their study at the end of the introduction. They have already carried out several studies in this area (as have others), so they must state what additional/unique contribution the present study makes.

The following text was added:

Serpentine products, suitable for obtaining magnesia-silicate reagent, were studied. These serpentines are included in the mining waste and their use at the large scale will contribute to the solution of two problems - the utilization of serpentine-containing wastes and the purification of aqueous solutions from metals. Granular magnesium silicate reagent made from thermally activated serpentine minerals is a promising material for purification of highly concentrated solutions. To obtain granular materials, a necessary condition is the presence of astringent properties, which will differ significantly for serpentines with different structures.

The conditions for the serpentines’ thermal activation are determined in accordance with the requirement to obtain materials with the highest possible degree of transformation of a serpentine mineral into an active metastable phase.

Since the type of the precursor magnesia-silicate binder directly affects the quality and properties of the final product [6]. The purpose of this work is to study the influence of data were obtained on the effect of the structure of the initial serpentine minerals on the hydration process of thermally activated serpentines and the strength of the binder based on them.

  1. Line 112: The authors should state how the solutions and dried precipitates were analysed.

The solutions and dried precipitations were analyzed for silicon and magnesium content by atomic emission spectrometry on an ICPE-9000 instrument (Shimadzu, Japan).

  1. Section 2.2.4: The exact water: binder ratio for each sample should be tabulated in this section.

Data of the water: binder ratio data was added

Line 121 Water demand of powders (solid/liquid ratio) for the plastic mixture obtain ranged from 0.37 (SLK, SAP) to 0.40 (SLH) and 0.44 (SCH), which corresponds to the normal density of cement grout.

  1. Section 3.1: The authors must provide a more comprehensive interpretation of the thermal events depicted in the DSC data shown in Figure 1. They should provide a clear account of all of the observed thermal events prior to making comparisons among the minerals. The data for DSC, thermogravimetry, XRD and FTIR should be presented consecutively to enable these data to be interpreted and explained more comprehensively.

The DSC results for serpentines are well documented in publications. In our studies, an important indicator that can be obtained on the basis of DSC data is the ratio between the main endothermic and exothermic peaks, which is described in the text of the article. At the beginning of the article, it was possible to cite the XRD results of only the initial samples, however, these data do not need to be described separately; it is advisable to compare them with the XRD of thermally activated and hydrated thermally activated samples, therefore they are given after the thermal activation conditions were chosen based on ANC. FTIR results are reported only for thermally activated chrysotile to confirm the presence of disordered forsterite in the sample. Based on XRD, it can be argued that forsterite is also present in the rest of the thermally activated samples.

  1. Table 2: The authors should explain the observed trends in acid neutralisation capacity in terms of the composition of the original and heat-treated minerals.

It was not indicated in the text of the article, but the ANC calculation for the correct comparison of samples with different amounts of water was calculated on the calcined residue (taking into account LOI 1000). Thus, the calculation is carried out for the same chemical composition. Clarification added to section 2.2.1:

The acid-neutralizing ability (reactivity) of heat-treated serpentines was calculated as the difference between the initial and remaining acid amounts [19], the formula for calculating ANC is given below. The result obtained is converted to the calcined residue (taking into account LOI 1000). The conversion to the calcined residue makes it possible to compare samples with different amounts of water, which remains in the structure of the serpentine mineral after calcination.

ANC = ,

ANC - acid-neutralizing capacity, meq/g,

C0 -- hydrochloric acid solution initial concentration, mol/L,

C1 - hydrochloric acid solution concentration after interaction with serpentine, mol/L,

m - serpentine sample portion, g,

V - volume of acid solution, l,

LOI  - loss on ignition at 1000℃, wt.%

  1. Lines 175-179: This section is confusing. The authors should more fully explain the information given in Lines 175-179. What is meant by ‘activity’? To what reaction do the authors refer when they discuss ‘activation energy’?  

Activity (reactivity) of serpentine samples correlates with ANC (see 2.2.1), ANC is called activity in this paragraph. In the Lines 175-179, activity was replaced with reactivity (the term is as in the publication [18]). In [22], the activation energies of serpentine dehydroxylation are given, i.e., reactions of water loss on heating.

  1. Figure 2: The authors should carry out full refinement on the XRD data for the original and heat-treated mineral samples to give a semiquantitative analysis their phase compositions and total degrees of crystallinity. They must also give the numbers of the powder diffraction files used to identify the phases.

For samples of antigorite and chrysotile, binder data are well interpreted based on such characteristics as ANC, degree of activation,and presence of binder precursor. For lizardite, the data were difficult to interpret, so these samples were studied in detail. In particular, the content of the crystalline phase was determined. Numbers of phase cards are added to the text of the article.

  1. Line 196 onwards: The FTIR spectra are missing. The authors must present and fully interpret the FTIR data.

The DSC results for serpentines are well documented in publications. In our studies, an important indicator that can be obtained on the basis of DSC data is the ratio between the main endothermic and exothermic peaks, which is described in the text of the article. At the beginning of the article, it was possible to cite the XRD results of only the initial samples, however, these data do not need to be described separately; it is advisable to compare them with the XRD of thermally activated and hydrated thermally activated samples, therefore they are given after the thermal activation conditions were chosen based on ANC. FTIR results are reported only for thermally activated chrysotile to confirm the presence of disordered forsterite in the sample. Based on XRD, it can be argued that forsterite is also present in the rest of the thermally activated samples.

  1.  Table 4: What is the origin of the theoretical acid neutralisation capacity of the minerals given in Table 4?

The “ability” was replaced by the “capacity” (also in the Title of the article)

Theoretical acid neutralization capacity was calculated from the magnesium content in heat-treated samples. An error was made in Table 4 - the theoretical ANC value is determined not by the uncalcined residue, while the experimental ANC is calculated for the calcined residue. The bug has been fixed, the data in the columns “Theoretical, meq/g” and “Activation degree, %” has been changed in the table.

Data for original samples from Table 4 has been removed as they are not discussed in the article.

Changes have also been made to paragraph 216-220.

  1. Table 4: Two columns of figures are given under the heading ‘Leaching degree’. Presumably these correspond with the respective leaching of MgO and SiO2. The table must be corrected to reflect this.

Corrected

  1. The discussion section is missing from the manuscript. The authors have presented and interpreted their data (in a confusing order), but they have failed to generate a discussion of their findings in the context of existing studies. They should also highlight the technical implications of their findings in the discussion, along with any practical applications of the thermally activated minerals and their binders.

The new Discussion section includes quantitative indicators (as recommended by Reviewer 2). The conclusion is changed, practical application of the results is stressed.

Reviewer 2 Report

The following items should be revised:

In the introduction part of the paper, please believe in clarifying the original point of this paper. At present, the original point of this article is not clear.

In the conclusion section of the paper, please include more details. The current conclusions only include qualitative analysis results, and more quantitative details should be added.

The bibliography contains a lot of Russian literature and should be replaced by the relevant English literature.

Figure 4 is not clear. Need to improve the clarity of this image。

The English of the article needs to be improved.

Author Response

Dear reviewer,

thank you very much for your valuable comments.

We have substantially revised the manuscript, especially abstract, Discussion and Conclusion sections, as well as English grammar.

The following items should be revised:

In the introduction part of the paper, please believe in clarifying the original point of this paper. At present, the original point of this article is not clear.

The following text was added:

On the other hand, thermally activated serpentines in the form of magnesia-silicate reagent were proposed to be used for water purification from heavy metals [10]. The influence of the structure of serpentine minerals on the activity of the reagent was considered [11]. It was revealed that the degree of serpentine transformation during firing decreased in the raw chrysotile – lizardite – antigorite. Method for determining the optimal firing conditions for the serpentines with high iron content was proposed [12]. The features of interaction of the reagent with solutions of heavy metals were determined; the interaction of thermally activated serpentinite with solutions of copper and nickel sulfates was considered [13].

Serpentine products, suitable for obtaining magnesia-silicate reagent, were studied. These serpentines are included in the mining waste and their use at the large scale will contribute to the solution of two problems - the utilization of serpentine-containing wastes and the purification of aqueous solutions from metals. Granular magnesium silicate reagent made from thermally activated serpentine minerals is a promising material for purification of highly concentrated solutions. To obtain granular materials, a necessary condition is the presence of astringent properties, which will differ significantly for serpentines with different structures.

The conditions for the serpentines’ thermal activation are determined in accordance with the requirement to obtain materials with the highest possible degree of transformation of a serpentine mineral into an active metastable phase.

The purpose of this work is to study the influence of data were obtained on the effect of the structure of the initial serpentine minerals on the hydration process of thermally activated serpentines and the strength of the binder based on them.

In the conclusion section of the paper, please include more details. The current conclusions only include qualitative analysis results, and more quantitative details should be added.

Quantitative indicators are described in the new Discussion section. The conclusion is changed, emphasis is placed on the practical application of the results obtained

The bibliography contains a lot of Russian literature and should be replaced by the relevant English literature.

We have replaced references 10, 13, 25 and 26 with papers by the same authors in English. We considered it undesirable to remove references to the earliest works on the formation of the metastable phase and on serpentine cements (7,8,9). It is also difficult to replace the reference to the work with the justification for the legality of using the ANC indicator, this material is published only in Russian. The link to this publication can be removed without replacing it with another publication, if the editor considers it necessary to remove it from the text of the article.

Figure 4 is not clear. Need to improve the clarity of this image

Unfortunately, we are unable to improve the quality of this drawing.

The English of the article needs to be improved.

The English grammar was corrected.

Reviewer 3 Report

Dear authors,

I think that your manuscript is interesting and suitable for this journal. It is worthy for publication after a moderate revision since I have some recommendations and comments presented below:

Line 51- 52.".thermall activated "...Please add "or mechanically" and recent related  reference

Figure 2: How are you sure about vermiculite detection and not another clay mineral? Have you done clay fraction analysis? Please clarify or add data. Please give the explanation of "P" abbreviation in Xrd pattern of Figure 2 . Have you detect any Ca-plagioclase? Moreover , add the reference pattern numbers from your database,  in case of the three serpentine minerals

Line 189: I think they are not poorly crystallized but remained crystals from the initial bulk composition of rock before alteration, please replece.

Line 241: Similarly remove the word "amorphous"

Lines 243-247 What about the Mg contributions from the forsterite and pyroxene (enstatite) minerals by the leaching. Please refer something. Good luck Kind regards

Author Response

Dear reviewer,

thank you for your valuable comments. We have substantially edited the body of the manuscript.

The answers to the comments are below:

Line 51- 52."thermall activated ".. Please add "or mechanically" and recent related  reference

Text at the end of Line 51-55 with links to works on mechanical activation of serpentines was added:

The most effective pre-treatment method for serpentine is thermal activation [Mech1], but mechanical activation is also used. Mechanically activated serpentine minerals can be used to purify solutions from metals, such as copper [Mech2].]

Figure 2: How are you sure about vermiculite detection and not another clay mineral? Have you done clay fraction analysis? Please clarify or add data.

Link to database was added, vermiculite card number 00-060-0341

Please give the explanation of "P" abbreviation in Xrd pattern of Figure 2 .

Caption to the Figure 2 was added “P - pyroxene (enstatite)”

Have you detect any Ca-plagioclase? Moreover , add the reference pattern numbers from your database,  in case of the three serpentine minerals

Line 141 - pattern numbers from database were added.

Line 189: I think they are not poorly crystallized but remained crystals from the initial bulk composition of rock before alteration, please replece. Line 241: Similarly remove the word "amorphous"

When interpreting the results, we rely on the concepts developed in the works carried out under the guidance of N.O. Zulumyan. The formation of forsterite and enstatite phases during the roasting of serpentine minerals is considered in detail in the works cited in the references.

Lines 243-247 What about the Mg contributions from the forsterite and pyroxene (enstatite) minerals by the leaching.

Pyroxenes and olivine are slightly soluble under the experimental conditions described in this article. In addition, they do not undergo changes during firing, and do not affect the processes of thermolysis of serpentinites described in the article and the subsequent processes of hydration and hardening of the obtained materials.

Round 2

Reviewer 1 Report

The authors have addressed some of the concerns that were raised during the review. It is still my opinion that the data could be better organised and more fully interpreted, and also that a robust discussion in the context of existing studies is lacking. I think these factors may limit the appeal of this study and its potential to be cited by other researchers. Nonetheless, if the authors are satisfied with quality of their scholarship, I recommend that the manuscript be published subject to moderate copy editing of the language and formatting.

I have no confidential comments for the editors. 

Author Response

The authors express their deep gratitude for the editing of the article, as well as for the condescending attitude towards the shortcomings. We agree that the article could be improved. At the same time, we believe that our results are of interest to those who develop methods for the practical application of serpentines. We hope that our data will contribute to solving the important problem of serpentine waste disposal.

Reviewer 3 Report

Dear authors,

your revised manuscript has been improved ant it can be published in this journal. Nevertheless, I consider that it is not accepted to refer amorpous forsterite or enstatite in case of  some peaks ( low or broad) are indicated in the XRD patterns: They are micro to nano crystals in an amorphous phase. 

Are you sure that the abbreviation O is needed in Figure 2? Do you refer to another olivine mineral except of forsterite (a solid solution of  Mg,Fe olivine? It is not clear, please check or remove it 

Kind regards

Author Response

The authors express their deep gratitude for the editing of the article, as well as for the condescending attitude towards the shortcomings.

We agree that the variant proposed by you for denoting the phases of forsterite and enstatite, which are formed during the roasting of serpentine minerals at a temperature below the temperature of the exothermic effect, is more correct. Corrected the text accordingly.

The phase, which is designated as olivine, was originally present in the source material. This is not forsterite, which was formed during the destruction of the serpentine mineral, it is precisely olivine, the presence of which is reflected in Figure 2.